# Neurotrophic Factors as Regenerative Therapy for Neurodegenerative Diseases: Current Status, Challenges and Future Perspectives

**DOI:** 10.3390/ijms24043866

**Published:** 2023-02-15

**Authors:** Yousra El Ouaamari, Jasper Van den Bos, Barbara Willekens, Nathalie Cools, Inez Wens

**Affiliations:** 1Laboratory of Experimental Hematology, Vaccine and Infectious Disease Institute (Vaxinfectio), University of Antwerp, Universiteitsplein 1, B-2610 Antwerpen, Belgium; 2Department of Neurology, Antwerp University Hospital, B-2650 Edegem, Belgium; 3Neurology, Translational Neurosciences, Born Bunge Institute, Faculty of Medicine and Health Sciences, University of Antwerp, B-2610 Antwerpen, Belgium; 4Center for Cell Therapy and Regenerative Medicine (CCRG), Antwerp University Hospital, Drie Eikenstraat 655, B-2650 Edegem, Belgium

**Keywords:** neurotrophic factors, regenerative therapy, neurodegenerative diseases

## Abstract

Neurodegenerative diseases, including Alzheimer’s disease (AD), Parkinson’s disease (PD), Huntington’s disease (HD), multiple sclerosis (MS), spinal cord injury (SCI), and amyotrophic lateral sclerosis (ALS), are characterized by acute or chronic progressive loss of one or several neuronal subtypes. However, despite their increasing prevalence, little progress has been made in successfully treating these diseases. Research has recently focused on neurotrophic factors (NTFs) as potential regenerative therapy for neurodegenerative diseases. Here, we discuss the current state of knowledge, challenges, and future perspectives of NTFs with a direct regenerative effect in chronic inflammatory and degenerative disorders. Various systems for delivery of NTFs, such as stem and immune cells, viral vectors, and biomaterials, have been applied to deliver exogenous NTFs to the central nervous system, with promising results. The challenges that currently need to be overcome include the amount of NTFs delivered, the invasiveness of the delivery route, the blood–brain barrier permeability, and the occurrence of side effects. Nevertheless, it is important to continue research and develop standards for clinical applications. In addition to the use of single NTFs, the complexity of chronic inflammatory and degenerative diseases may require combination therapies targeting multiple pathways or other possibilities using smaller molecules, such as NTF mimetics, for effective treatment.

## 1. Introduction

Neurodegenerative diseases of the central nervous system (CNS), such as multiple sclerosis (MS), Alzheimer′s disease (AD), Parkinson′s disease (PD), Huntington′s disease (HD), amyotrophic lateral sclerosis (ALS), and in acute cases, spinal cord injury (SCI), are still incurable and have high individual and societal costs [1,2,3]. PD and AD are the most common neurodegenerative diseases. As the world′s population ages, the prevalence of AD and PD is rapidly increasing. It is estimated that 50 million people worldwide suffer from neurodegenerative diseases, and this number will rise to 115 million by 2050 [4].

Unfortunately, currently available treatment options are inadequate to halt neurodegenerative processes [5,6]. Moreover, our understanding of the pathogenic processes and the consequent development of effective treatments is significantly complicated by the complexity of the mechanisms associated with neuronal loss and the conflicting physiological causes of these diseases. Furthermore, the difficulty in addressing widespread neuronal cell death, combined with the enormous limitations for the vast majority of drugs not to cross the blood–brain barrier (BBB), further complicates the treatment of these diseases [7,8].

From an evolutionary point of view, the nervous system would be able to protect itself from any injury [9]. In the early 20th century, pioneering work by Tello and Cajal demonstrated that the CNS has the ability to regenerate itself after injury [10,11,12]. In recent years, researchers have accumulated detailed in vitro and in vivo mechanistic evidence supporting the idea that an innate self-maintenance program is activated in the brain, not only during inflammatory and degenerative diseases, but also in healthy individuals [11,13,14]. These observations support the idea that chronic inflammatory and degenerative disorders of the brain can be the result of defective repair mechanisms, rather than uncontrollable pathogenic events [11,15,16,17]. We can, therefore, subscribe the idea that failure of molecular and cellular mechanisms sustaining the “brain-repair program”—which can be considered as an intrinsic part of the physiological activities of the brain—might be, at least partially, a cause of neurodegenerative diseases [11,18]. Therefore, research into the molecular and cellular events sustaining intrinsic brain-repair mechanisms and a better understanding of why they fail over time in chronic disorders might provide an attractive conceptual framework, in which new and efficacious therapies for neurodegenerative diseases can be developed.

Neurotrophic factors (NTFs) and their receptors play a crucial role in neural cell maturation and proliferation. NTFs regulate the development and survival of neurons, and they appear to be involved in the endogenous neuroprotection of different neurons. Several studies have reported that NTFs, particularly glial cell-derived neurotrophic factor (GDNF), ciliary neurotrophic factor (CNTF), brain-derived neurotrophic factor (BDNF), nerve growth factor (NGF), neurotrophin-3 (NT-3), and neurotrophin-4/5 (NT-4/5), act regeneratively in different animal models [19,20,21,22,23,24,25,26,27,28,29,30,31,32,33,34,35,36,37,38,39,40,41,42,43,44,45,46,47,48,49,50,51,52,53,54,55] and patients [56,57,58,59,60,61,62,63,64,65,66,67,68,69,70,71,72,73] with neuroinflammatory and neurodegenerative diseases. Consistent with their known role in maintaining neuronal homeostasis, these NTFs, with regenerative properties, have been proposed as novel therapies for several neuroinflammatory and neurodegenerative diseases [74,75,76]. In this review, we provide an overview of the various and known NTFs described in the literature with their effects in the CNS. As well, we summarize the different approaches where NTFs have been administered via direct delivery or delivery through a vehicle, such as stem and immune cells, viral vectors, and biomaterials, into animal models or in patients suffering from a neurodegenerative disease.

## 2. Functions and Mechanisms of Neurotrophic Factors in Neurogenesis and Brain Repair

Glial cell-derived neurotrophic factor (GDNF) was originally isolated from the supernatant of a rat glioma cell line and found to have pronounced effects on the survival of midbrain dopaminergic neurons [77,78,79]. GDNF has further critical roles outside the nervous system in the regulation of kidney morphogenesis and spermatogenesis [80]. In the case of potential therapy for neurodegenerative diseases, GDNF has a relatively high specificity for dopaminergic neurons and, thus, has significant potential for the treatment of PD, which is mainly characterized by progressive depletion of dopaminergic cell populations in the midbrain [79]. Subsequently, GDNF was also shown to have trophic and protective effects on noradrenergic neurons in the locus coeruleus and on peripheral motor neurons, giving hope for its therapeutic potential in HD and ALS [24,25,35,51,81,82,83]. Translational research has focused mainly on the treatment of PD, where there has been reason for both celebration and caution [27,79,84,85,86,87,88,89,90,91,92,93]. A recent review by Manfredsson et al. [94] has highlighted that the therapeutic mechanism of action of GDNF is not fully well-defined, and that the degenerating brain of PD may be resistant to the neuroprotective potential of these proteins. The lack of clarity on the mechanism of action of GDNF may cause problems in appropriate model selection for preclinical therapeutic studies [94].

A second interesting NTF is the ciliary neurotrophic factor (CNTF), which is a member of the interleukin-6 family of cytokines. It has potent effects on the development and maintenance of the nervous system, as well as on cardiomyocytes, osteoblasts, immune cells, adipocytes, and skeletal muscle cells [95,96]. CNTF has been found to affect motor neuron survival in vitro, during development, after injury to motor neuron systems, and in genetic models of motor neuron degeneration [57], providing a rationale to develop CNTF as a treatment for ALS [56,57,62,66,67,68,69] and SCI [49,52], in which ventral motor neuron degeneration is extensive [57]. A drawback for CNTF administration is that it protects motor neurons from degenerative disease and injury, but also has some side effects, such as severe weight loss, hyperalgesia, coughing, muscle cramps, and pain [97]. Therefore, CNTF-related therapeutics will need to be designed to specifically target receptor mechanisms that protect motor neurons [98].

Apart from GDNF and CNTF, other known factors are the neurotrophins. This group consists of four members that share a common ancestral gene and have a similar structure. In the CNS, brain-derived neurotrophic factor (BDNF) is the major neurotrophin because of its abundant expression of tropomyosin receptor kinase B, also known as tyrosine receptor kinase B (TrkB) [99]. Studies of disease models of AD, in which BDNF was increased by using, for example, a lentivirus that expressed BDNF, showed that this factor is essential for multiple functions during adulthood, such as proper memory acquisition, memory retention, and cholinergic innervation [100,101]. BDNF is decreased within the brains, serum, and cerebro-spinal fluid (CSF) of patients with mild cognitive impairment and AD [101]. Also, low BDNF secretion in the serum of MS patients may be related to reduced neuroprotection [102,103]. As a result, low BDNF levels are expected to diminish the potential for remission in MS patients and induce the progressive phase of the disease [104]. To date, the potential beneficial effect of BDNF has been explored in several neurodegenerative and inflammatory diseases, such as animal models of AD, SCI, MS, and HD [21,54,99,105,106,107,108]. 

In the peripheral nervous system (PNS), nerve growth factor (NGF) is the dominant neurotrophin, which interacts on sympathetic and sensory neurons. In the CNS, NGF specifically provides trophic support to cholinergic neurons of the basal forebrain (BFCNs) that express TrkA (Figure 1), which would make it specifically interesting for AD [63,64,65,109,110]. NGF and its receptors, TrkA and p75, are known to play a bidirectional role between the immune and nervous systems. Recently, it has been extensively discussed that NGF plays a dual role in both anti- and pro-inflammatory response [111]. Moreover, cytokines, such as IL-1β, TNF-α, and IL-6, induce the overexpression of NGF [112].

Finally, neurotrophin-3 (NT-3) and neurotrophin-4/5 (NT-4/5) also have promising potential, albeit less studied than their counterparts. NT-3 is the third neurotrophic factor of the neurotrophin family, and, through activation of its tropomyosin-related kinase receptor C (TrkC) (Figure 1), it can modulate neuronal survival, support the differentiation of neurons, and stimulate the growth [113] and differentiation of new neurons and synapses [45]. Although this neurotrophin seems less popular, interesting in vivo studies have been done in various neurodegenerative diseases [39,45]. 

NT-4, also known as neurotrophin-5 (NT-5), is a neurotrophin that primarily signals via the TrkB receptor tyrosine kinase (Figure 1). The neurotrophins BDNF and NT-4 both bind to and activate TrkB receptors; however, they mediate different neuronal functions. The molecular mechanism of how TrkB activation by BDNF and NT-4 results in different outputs is not yet known. NT-4 is the least studied member of the neurotrophin family [53,114,115,116,117].

Unfortunately, the exact mechanism of NTFs is not yet fully understood. Nevertheless, research already reported the different NTF receptors and unravelled the pathways they activate to ensure the maintenance of cell growth, survival, development, and differentiation. BDNF, NGF, NT-3, and NT-4/5 bind to two families of receptors, namely, the tropomyosin kinase (Trk) receptors with high affinity, and with low affinity to the p75 receptor (Figure 1) [74,75,118]. Their actions are dependent on binding to the transmembrane receptor systems. Neurotrophins preferentially bind to specific receptors: NGF binds to TrkA, BDNF and NT-4 to TrkB, and NT-3 to TrkC [119]. However, there are a number of promiscuous interactions. All four neurotrophins can bind to the p75 receptor, and the association of p75 with Trk receptors can regulate the affinity of Trk receptors for each respective neurotrophin, allowing more control over ligand-receptor interactions within this system [120]. The Trk receptor binds with high affinity with NTFs to promote cell survival via phospholipase C-γ (PLC-γ), phosphoinositide-3-kinase (PI3K), and mitogen-activated protein kinase (MAPK) pathways that induce differentiation and survival via transcriptional events (Figure 1, green arrows). The MAPK pathway may be involved in ureteric branching during nephrogenesis and neurite outgrowth in the nervous system, but it also contributes to neuronal survival. The PI3K pathway is crucial for both neuronal survival and neurite outgrowth. The PLC-γ pathway regulates the intracellular level of Ca^2+^ ions by increasing the level of inositol (1,4,5) trisphosphate. Binding of NTFs to the low-affinity p75 receptor activates cell death via the JNK pathway. Activation of the JNK pathway similarly controls activation of several genes, some of which promote neuronal apoptosis. Neurotrophins are known to have a wide range of roles in the development and function of the nervous system. The characterisation of their receptors—the Trk receptor and p75 receptor—has significantly advanced research and enabled the characterisation of signalling pathways and the first steps to relate individual signalling pathways to specific developmental or functional roles of neurotrophins [119].

Today, we have learned to which receptors the various NTFs bind and what signalling pathways they activate. For instance, binding of CNTF to the CNTFRα receptor and two subunits, GP130 and leukaemia inhibitory factor (LIFRβ), activates the Janus kinase/signal transducer, an activator of transcription (JAK-STAT), MAPK, and PI3K pathways. The JAK-STAT pathway is associated with cell growth, survival, development, and differentiation (Figure 1, violet and blue arrows). Binding of GDNF to the GFRα receptor and tyrosine kinase RET receptor stimulates PLC-γ, MAPK, and PI3K (Figure 1, yellow arrows). RET activates various intracellular signalling cascades, which control cell survival, differentiation, proliferation, migration, chemotaxis, branching morphogenesis, neurite outgrowth, and synaptic plasticity [120]. Akt controls the activities of several proteins important for promoting cell survival, including substrates that directly regulate the caspase cascade, such as Bcl-2 agonist of cell death (BAD). Phosphorylated BAD prevents its pro-apoptotic activity (Figure 1, red inhibitory arrow). These different signaling pathways, which are activated by NTFs, work together to ensure normal neuronal function and to prevent neuronal cellular death (Figure 1).

## 3. Delivery of NTFs’ and Associated Challenges

### 3.1. Administration of NTF by Direct Infusion in the CNS

Various techniques have been used to get NTFs into the brain. The best known technique is direct intracerebroventricular (ICV) infusion. In particular, recombinant human (rh) GDNF and ^125^Iodine-labelled GDNF (^125^I-GDNF) have been shown to diffuse into the deep brain structures of rats [79,86], not only to significantly increase striatal and nigral dopamine (DA) levels, but also to increase hypothalamic DA levels, which could explain the decreased food and water consumption and body weight observed in in vivo experiments [87,88]. ICV injection of GDNF into 6-hydroxydopamine (6-OHDA)-treated rats, an animal model of PD, also appears to result in improved locomotor performance [87,88]. Furthermore, the ICV delivery route seems suitable for therapies that need to reach the BFCNs. Early and progressive degeneration of BFCNs contributes substantially to cognitive impairments of AD. Since BFCNs extend their axons through the hippocampus and neocortex, NGF administered in the lateral ventricle can act on the TrkA receptor to transmit trophic support signals to BFCNs. This approach has been shown to be particularly effective in preventing loss of BFCNs in rodents associated with injury and ageing [110,122,123]. However, the small volume of the rodent brain compared to the human brain raises important questions about the applicability of this technique in clinical studies. Therefore, ICV injections were also performed in non-human primates [90,91,92]. GDNF has been shown to produce significant improvements in motor activity in 1-methyl-4-phenyl-1,2,3,6-tetrahydropyridine (MPTP)-treated rhesus monkeys, a model of PD [89,90], and improvements in motor impairment and reductions in l-dopa-induced dyskinesia in marmosets [91]. In an autoradiographic study of the distribution of ^125^I-GDNF administered in the lateral ventricles of rhesus monkeys with a MPTP lesion, GDNF was not found to diffuse readily into the putamen. This finding contrasts with similar studies in rodents [79,86], suggesting that the success of ICV infusion in rodents might be a product of the smaller diffusion distance within their brain [87,88]. Moreover, the ICV delivery route was associated with serious side effects [110], such as hyperinnervation of cerebral blood vessels [123], hypophagia [110,122], Schwann cell hyperplasia with sprouting of sensory and sympathetic neurons [124], neuropathic pain [110], and dyskinesia [89,90,91], providing profound contra-indications for the applicability in clinical trials. 

Because of these ICV-related side effects, the study by Tuszynski et al. [109] investigated whether intra-parenchymal infusion would be a well-tolerated way to administer NTFs to degenerating cholinergic neurons. In particular, intraparenchymal NGF infusion prevented degeneration of BFCNs, whereas glial responses were minimal in adult rats that underwent complete unilateral fornix transections, followed by intraparenchymal infusions of recombinant human NGF for a 2-week period. In addition, no apparent toxic effects of the infusions were observed, according to the researchers [109]. Other studies aimed to administer NTFs by a less invasive method. The group of Braschi et al. [21] tested whether intranasal (IN) administration of different concentrations of BDNF in AD11 transgenic mice, a model of AD, was able to rescue neuropathological and memory deficits. They found that IN administration of BDNF, but not with PBS, was adequate to completely rescue the performance of AD11 mice in both the object recognition test and the object context test. The strong improvement in memory performance in BDNF-treated mice was not accompanied by an improvement in AD-like pathology, amyloid-β (Aβ) load, tau hyperphosphorylation, and cholinergic deficiency [21]. Similarly, IN administration of NGF to Aβ peptide-expressing traumatic brain injury (TBI) rats, which are at risk of AD in later life, caused a marked reduction in Aβ42 deposits and restored motor and behavioural function [20]. Features such as non-invasive manipulations, rapid absorption rate, easy repetitive dosing, and reduction of non-target biodistribution make IN administration superior to the systemic and ICV routes of administration [19,20].

Finally, studies examined the effects of continuous intraputamenal administration of GDNF in both aged and MPTP-lesioned non-human primates [84,85,93]. Histological and biochemical analysis showed an increase in cell size and the number of dopaminergic neurons within the substantia nigra, as well as increased fibre density in the caudate nucleus, putamen, and globus pallidus. Primates with MPTP lesions showed improvements in the primate PD rating scale, whilst aged monkeys demonstrated improvement in general motor performance at high doses and increases in hand speed [84,85,93]. To assess the possible side effects of continuous administration of GDNF, a six-month toxicity study was conducted in rhesus monkeys. The results cast considerable doubt about the neuro-restorative potential of GDNF for the treatment of PD, given that they identified a number of pathological markers of toxicity, including reduced food intake and weight loss, meningeal thickening, and most concerning, multifocal cerebellar Purkinje cell loss [31]. Apart from the above-mentioned side effects, direct administration of NTFs into the brain also had some practical problems, such as invasiveness, BBB permeability [7,8,125], poor half-life, and rapid degradation [126]. This led to studies using cell therapy, where cells were modified to produce a specific protein.

### 3.2. Cells Modified to Express Neurotrophic Factors

During the last years, different cell types have been utilized to deliver NTFs to the injured sites. Mesenchymal stromal cells (MSCs) are described as adherent, fibroblast-like cells with prominent proliferation capacity [42,50,51,127]. Because of their low immunogenicity (low expression levels of major histocompatibility complex (MHC) class II), MSCs can survive after administration [128]. The existence of such capabilities makes MSCs a safe, tolerable, and efficient biological vector for the generation and delivery of therapeutic agents, such as NTFs, to the target sites [42,50,51]. Furthermore, different routes of administration were used to administer the modified MSCs, resulting in different outcomes. In a study by Suzuki et al. [51], human MSCs (hMSCs), derived from neonatal bone marrow aspirates which were modified to express GDNF, were administered intramuscularly as a "Trojan horse" to superoxide dismutase (SOD1)^G93A^ rats, a rat model of familial ALS, to deliver GDNF to the terminals of motor neurons and to skeletal muscle. hMSC-GDNF survived in the muscle, secreted GDNF, and significantly increased the number of neuromuscular connections and motor neuron cell bodies in the spinal cord in the mid-stage of the disease. Moreover, hMSC-GDNF significantly slowed down disease progression [51]. In addition, several improvements have been reported when CNTF- [52], NT-3- [53], and BDNF-modified [54] MSCs were administered directly into the spinal cord of SCI rats, such as improvement in behavioural scores, motor function, axonal regeneration, and neuronal survival [52,53], and restoration of diaphragm muscle function [54]. Positive results with MSCs expressing NTFs were also observed after intravenous (iv) administration. A remarkable recovery of neuronal function was observed and demyelination was significantly reduced in EAE mice: the cumulative clinical scores were significantly decreased, and the disease onset was statistically delayed, after iv MSC-CNTF [55] and MSC-BDNF administration [105]. Moreover, BDNF-expressing MSCs can also reduce striatum atrophy and increase neurogenesis in HD mouse models [22]. In summary, MSCs represent a promising tool for cell therapy. There is currently much interest in the use of MSCs for the treatment of neurodegenerative diseases. There are several studies using the innate trophic support of MSCs or increased support by NTFs, such as the administration of BDNF, CNTF NTF-3, or GDNF to the CNS to support damaged neurons, using genetically engineered MSCs as delivery tools. Biosafety could be a potential difficulty in cell therapies when using genetically engineered MSCs. The random integration of vectors with genes for neurotrophic or other factors may pose the risk of insertional integration. However, homologous recombination and targeted gene transfer are advancing rapidly.

Neural stem cells (NSCs) are also used as a NTF vector, resulting in several positive effects. NSCs are characterised as multipotent and self-renewing cells with the capacity to differentiate into mature neurons and neuroglia cells [23,24,25,26]. In a rodent model of cervical SCI, it was shown that GDNF-expressing human induced pluripotent stem cell-derived NSCs (hiPSC-NSCs) showed greater differentiation into a neuronal phenotype than unmodified hiPSC-NSCs [27]. Furthermore, several improvements were seen with NSCs expressing GDNF in SOD1^G93A^ ALS rats, when administered in the motor cortex [24] and in the spinal cord [25]. The results show improved survival, as well as enhanced proliferative and neuroprotective properties [24,25]. Moreover, human GDNF-expressing NSCs duly migrated to the disease site and integrated into the CNS after administration into the spinal cord of SOD1^G93A^ ALS rats [25]. In addition, it has been shown that GDNF-expressing NSCs administration in the lateral ventricle promotes axonal regeneration and remyelination in chronic EAE rats [26]. 

A number of studies have indicated that immune cells are also useful as therapeutic biosystems to deliver various molecules into target areas [28,29]. Among the subsets of immune cells, macrophages are the most suitable target cells, as they are activated soon after the onset of the inflammatory response, can cross the BBB, and move to sites of neuronal degeneration [28,29]. In this regard, the monocyte-macrophage lineage could represent an efficient cellular system to deliver NTFs at the site of injury within the CNS. To support this hypothesis, Biju et al. used ex vivo transduced bone marrow-derived macrophages to deliver GDNF [28]. Axonal regeneration and retention of tyrosine hydroxylase (TH+) neurons were observed in both the striatum and substantia nigra regions [28]. Moreover, GDNF-expressing macrophages could successfully cross the BBB and deliver GDNF into the neuro-generated DA neurons after systemic administration [29].

Finally, other cells, such as fibroblasts, were also used as vectors to deliver NTFs. Specifically, fibroblasts modified to express BDNF were inoculated into SCI sites in rats, and these caused regenerative and sprouting responses at the sites of injury [106,107,108]. Similarly, genetically modified baby hamster kidney (BHK) cells and primary cells expressing NGF showed that they were able to rescue cholinergic function in damaged neurons in ageing models of both rodents and non-human primates [129,130,131]. More interestingly, the implanted cells maintained NGF secretion for at least 8 months in primate brains and did not cause the adverse side effects observed in studies with direct administration [132,133,134].

To date, research advances in cell-based therapies offer promising methods for treating neurodegenerative diseases. Although much work remains to be done, the increasing focus on preclinical studies and the recent translation of some of these therapies into clinical trials have paved the way for further progress. The use of modified cells expressing NTFs is likely to play a key role in future clinical strategies to treat neurodegenerative diseases by replacing dysfunctional neurons and providing neuroprotective functions. As mentioned earlier, a potential drawback that remains today is the biosafety.

### 3.3. Viral Delivery of Neurotrophic Factors

Viral vector-mediated gene delivery might be a more optimal approach instead of the techniques that have been previously described. Virus administration would permanently alter the cells’ ability to make its own NTF, requiring a single injection at the site of administration, rather than multiple injections [82,83,135,136], and eliminating the cumbersome cell preparation associated with the cell transfer technique [30,32,33,34,36,82,83,135,136,137].

Nakajima et al. [30] reported that injection of adenovirus (AV)-BDNF into bilateral sternomastoid muscles transferred vectors to the damaged sites, via retrograde transport using spinal accessory motor neurons, in SCI rats. The AV-BDNF was able to reach the spinal cord and reduce apoptotic signalling in neurons and oligodendrocytes [30]. Likewise, the application of retrograde AV-BDNF in bilateral sternomastoid muscles of chronically compressed SCI mice led to the recovery of oligodendrocyte progenitors and neurofilament expression via the axons of spinal accessory nerves [32]. However, there are some drawbacks using AV vectors, including immunogenicity, replicability, and the small insertion size of the vectors [30,32].

To date, adeno-associated virus (AAV)-mediated gene transfer of GDNF has been used and evaluated in a number of studies in rodents and primates, particularly for PD [136], HD [82,83], and SCI [33]. Eslamboli et al. [136] showed that unilateral intrastriatal injection of AAV-GDNF, resulting in the expression of high levels of GDNF in the striatum, induced a significant bilateral increase in tyrosine hydroxylase protein levels and DA turnover in a 6-OHDA lesion in marmosets. In addition, AAV-GDNF-treated rats scored better on a blinded semi-quantitative neurological scale compared to rats receiving the control AAV- Green Fluorescent Protein (GFP), which was supported by histological analyses [83]. Interestingly, Fouad et al. [33] reported that rats, with complete thoracic SCI, that received combined treatment, including self-complementary AAV-BDNF and NT-3 administration in the spinal cord, showed not only improved axonal regeneration, but also improved motor function of the hind limbs [33]. AAV vectors offer many of the same advantages as AV vectors, including a wide host-cell range and a relatively high transduction efficiency. In addition, AAV vectors do not express their own proteins and, therefore, would not elicit an immune response, making the technique even more attractive. However, the major drawback is the limited cloning capacity of the vector, which restricts its use in the gene delivery of large genes [33,82,83,136]. 

Next to AV- and AAV- mediated NTF delivery, viral delivery of GDNF by lentivirus (LV) reversed motor deficits and prevented nigrostriatal degeneration in MPTP-treated monkeys [137]. The delivery of LV expressing GDNF to AD mice models enhanced learning and memory function, while simultaneously improving the cognition capacity [34]. In addition, the group of Pereira de Almeida et al. [138,139] conducted two studies using tetracycline-regulated LV-mediated delivery of CNTF in a quinolinic acid (QA) rat model of HD. The 2001 study [138] showed that the extent of striatal damage was significantly reduced in the CNTF-treated rats, and the volume of the lesion was significantly reduced [138]. In 2002, they reported CNTF′s dose-dependent effects [139]. Remarkably, LV-based administration has numerous advantages, such as long-term transgene expression, low inflammation rate, and large-size gene insertion [35,36,140]. Despite these advantages, in some cases, oncogenic mutation may occur after integration of the LV gene into the host cell genome. This is cited as the main concern of safety in in vivo conditions.

### 3.4. Biomaterials to Deliver Neurotrophic Factors

Several of the above-mentioned strategies to deliver NTF to the site of injury in the spinal cord or brain, such as direct delivery, genetically engineered cells, and viral vectors, have a number of drawbacks, including viral vector spread beyond the target area, uncontrolled transgene expression, and immune rejection of transplanted cells. Therefore, there is a growing interest in using biomaterials as vehicles to deliver NTFs. Natural biomaterials are biocompatible, biodegradable, have remodelling advantages and a lower toxicity rate [141], while synthetic biomaterials have a more favourable mechanical and thermal resistance, no immune response capacity, and can be produced on large scales [37,38].

A recent study by Zhijiang et al. [141] used the natural biomaterial methylcellulose (MC), combined with hyaluronic acid (HAMC) hydrogel modified with the peptide KAFAK-LAARLYRKALARQLGVAA (KAFAK) and BDNF. They injected these into a lesion area of SCI rats and showed that locomotor function and axonal regeneration improved 8 weeks after SCI [141]. A similar study with NT-3 also showed that HAMC could release NT-3 for 28 days. The persistence of NT-3 in the target areas confirmed the regeneration and expansion of axons, without induction of the astroglial response, which can cause an inflammatory reaction [39]. Furthermore studies have used other natural bio-materials, such as bioactive scaffolds, to create a microenvironment conducive to endogenous regeneration of neuronal tissue in the SCI site. In particular, gelatin sponge scaffold, silk fibroin, chitosan, or a more developed multichannel nanofibrous gelatin scaffold have been used. These scaffolds were integrated into NT-3, with or without NSCs [44], adipose-derived stem cells [43], or MSCs [45,142]. The in vivo experiments have significantly improved neuronal differentiation, synaptic connection, and axonal remyelination, with reduced local inflammation at the SCI sites following bioactive scaffold implantation with NT-3. In addition the treatment has shown significant improvement in locomotor functionality [40,43,44,45,142].

Poly-lactide-co-glycolide (PLG) is one of the most frequently used synthetic biomaterials for drug delivery, due to its controlled and sustained release properties, low toxicity, and biocompatibility with tissue and cells [46,47]. PLG has been widely used as a material for spinal cord repair or peripheral nerve conduits [47]. Khalin et al. found that iv injection of poloxamer 188 (PX)-coated PLG nanoparticles with BDNF (PLG-BDNF) in TBI mice restored cognition and showed that this system is eligible to cross the BBB and deliver BDNF into the brain of the TBI model [38]. Furthermore, several studies with PLG-BDNF in animal models of SCI observed robust axon growth and remyelination 6 months after initial injury [39,47,48]. These positive findings of PLG-BDNF were not confirmed with CNTF. The latter would not be sufficient in vivo to promote oligodendrocyte remyelination in the glial-depleted environment of unilateral ethidium bromide lesions [49]. Similar to the PLG-BDNF results in SCI rats, poly N-isopropylacrylamide (PNIPAAm) with BDNF improved the axonal regeneration in SCI rats [37]. Finally, intrathecal infusion of N-terminal pegylated (PEG) BDNF (PEG-BDNF) was also used in an attempt to increase NTF release [143]. The authors showed that the PEG-BDNF was able to reach the spinal cord and that its expression was induced in that area. However, they could not observe an improved axonal response or recovery of motor function, which suggests that the amount of BDNF was insufficient [143].

As mentioned earlier, most NTFs have difficulties passing through the BBB and are, therefore, delivered directly into the brain in animal models and some clinical trials with patients using expensive and risky intracranial surgery [70,71,72]. The efficiency of delivery and the poor distribution of some NTFs in the brain are considered the main problems behind their modest effects in clinical trials. There is a great need for NTFs that can be administered systemically to avoid intracranial surgery. Nanoparticles (NPs) can be used to stabilise NTFs and facilitate their transport through the BBB [144]. For example, one study used plasmid DNA NPs encoding human GDNF (pGDNF) that were administered IN to a rat model of PD [145]. The amphetamine-induced rotational behaviour was reduced, and dopaminergic fibre density and cell counts in the lesioned substantia nigra and nerve terminal density in the lesioned striatum were significantly preserved in rats given IN pGDNF [145].

## 4. Clinical Trials with Neurotrophic Factors

In addition to studies in animal models, there were also studies in humans, in which NTFs were used for the purpose of regeneration. The first clinical trials with NTFs in ALS patients applied systemic administration of CNTF, while the protein did not readily cross the BBB and consequently did not reach a detectable concentration in the central parenchyma [56,57,66,67,68,69]. Side effects, including inflammation and cachexia, have been recorded after systemic administration, which were severe enough to terminate phase II/III clinical trials with CNTF in ALS patients [56,57,66,67,68,69] (Figure 2). This led to the NTFs being administered directly into the brain in subsequent clinical studies. In particular, GDNF was administered by monthly bolus injections into the cerebral ventricles of PD patients. No beneficial clinical effects were seen, whereas side effects, such as nausea, loss of appetite, tingling, Lhermitte sign, intermittent hallucinations, and depression, were reported. In addition, there was no evidence of the restoration of dopamine fibers in the striatum [70,71]. Bolus injection into the parenchyma exposed the patient to a higher risk of tissue trauma and denied the clinician the means to finetune and optimize dose delivery (Figure 2). The clinical phase I safety trial of Nikunj et al. delivered GDNF directly into the putamen of five patients with PD [72]. Afterwards, they continued to follow these patients for two years and concluded that direct intraputamenal GDNF infusion in patients with PD is safe, can be tolerated for two years, and leads to significant symptomatic improvement [73]. Interestingly, the same group performed another randomized, controlled, blinded clinical trial in order to confirm the initial clinical benefits. However, this trial did not confer the predetermined level of clinical benefit to patients with PD, despite increased (18)F-dopamine uptake [58].

The macro-encapsulation technique was a more sophisticated method. This technique was first conducted with CNTF in rats and non-human primate models of HD [59,60]. In brief, BHK cells engineered to synthesize and release large amounts of NTF, such as CNTF, have been introduced into a tube formed by a semipermeable membrane. The pores of this membrane are sized so that proteins can cross freely, whereas larger proteins (e.g., antibodies) and cells cannot. Due to the positive results of this technique, reduced side effects, and the ability of BHK-hCNTF to protect neurons from degeneration and restore neostriatal function in animal models [59,60], the group of Bachoud-Le Âvi et al. [61] and Aebischer et al. [62] used this macro-encapsulation technique in a phase I study in ALS and HD patients (Figure 2). In particular, a capsule was introduced into the lateral ventricle of six patients with HD [61] and ALS [62], using stereotactic neurosurgery. No signs of CNTF-induced toxicity were observed. According to the results, this phase I study demonstrated the safety, feasibility, and tolerability of this gene therapy procedure, but the heterogeneous cell survival indicates the need to improve a more uniform response. Furthermore, no clinical benefit was observed in any of the treated subjects, which could partly be due to the limited diffusion of CNTF through the ventricular wall to the adjacent putamen [61,62], similar to the limited diffusion of GDNF after ICV injection in non-human primates [89,90,91,92].

Finally, some clinical studies have used cells or viral vectors to bring the NTFs into the brain. Mark Tuszynski′s team [63] surgically implanted autologous fibroblasts, which were modified to secrete mature human NGF, into the basal forebrain of eight early stage AD patients. The mean Mini-Mental Status Examination (MMSE) scores showed an average decrease of 51% over a 22-month period, and an even greater decrease over 6 to 18 months. Moreover, there were cognitive improvements, and post-mortem analysis confirmed that there was NGF expression in the cell grafts and that cholinergic axons showed outgrowth. Overall, this study presented the first clinical evidence that NGF administration can provide therapeutic benefit, without side effects usually associated with NTF administration, such as nausea, loss of appetite, tingling, hallucinations, and depression [63]. Because AAV serotype 2 (AAV2)-NGF vectors represent a more convenient and less expensive method of gene delivery and resulted in long-term gene expression in non-human primate brains [64], Tuszynski et al. conducted a second phase 1 clinical trial on 10 patients with AD (Figure 2). Here, AAV2-NGF was injected in vivo into the basal forebrain region, genetically modifying cells of the brain itself, rather than employing grafts of autologous cells, as employed in the phase 1 ex vivo study [65]. This study showed that responses to NGF persist for up to 10 years after gene transfer. No adverse pathological effects were observed over a 7-year period, supporting the safety and rationale for the expanded clinical programs underway in AD, PD, and other neurological indications [65].

## 5. Challenges and Future Perspective of the Use of NTFs in Neurodegenerative Diseases

Neurodegenerative diseases that cause acute or chronic damage to neurons and glial cells represent a major socio-economic burden and loss of quality of life for millions of patients and their families worldwide [3]. With an ageing population, the number of patients will further increase [4], creating an urgent need for therapeutic strategies that can reverse or stop the degenerative process. NTFs, as discussed in this review, are important factors in both development and adulthood, and each is required by certain subsets of neurons for optimal function. From the results, GDNF would be of particular interest for PD, due to its high specificity for dopaminergic neurons [84,85,91,93,124]. In addition, CNTF seems important, especially for ALS [56,57,62,66,67,68,69] and SCI [49,52], due to its potent effects on motor neuron survival, after injury to motor neuron systems and in genetic models of motor neuron degeneration. NGF specifically provides trophic support to cholinergic neurons of the BFCNs that express TrkA, which would make it of particular interest for AD [63,64,65,109,110]. The potential beneficial effect of BDNF has been studied in several neurodegenerative and inflammatory diseases, including animal models of AD, SCI, MS and HD [21,54,99,105,106,107,108]. As well, neurotrophin-3 (NT-3) and neurotrophin-4/5 (NT-4/5) also have promising potential, however, they have been less studied than their counterparts. Decreased levels of one or more of these proteins may be responsible for at least some of the symptoms of AD, PD, ALS, HD, and MS [78,101,103,104,146,147]. Therefore, these factors have been investigated as a potential neuro-healing therapy in preclinical and/or clinical studies (Figure 2). In particular, NTFs can be delivered via direct infusion, cells modified to (over)express these factors, viral delivery, or biomaterials (Figure 2).

There are strong arguments showing that an increase of NTFs-delivery to degenerating neurons could be a powerful way to restore neuronal function, but the delivery of these NTFs into the brain seems challenging [148]. In particular, diseases of the CNS are known to be difficult to treat because of the presence of the BBB, which makes it virtually impossible for large proteins and complex connections to enter the brain from the blood [149,150,151]. The possibility that NTFs can cross the BBB is quite controversial [148]. For example, some authors state that it is not clear whether BDNF can easily pass the BBB [152], whereas others indicate that BDNF is able to do so [153]. Molinari et al. [153] published a recent paper on the possibility of using exogenous BDNF as a therapeutic approach in neurodegenerative diseases. His work showed, in in vitro experimental models, that a low BDNF dose can cross both the intestinal and BBB barrier [153]. An alternative way and more recent technique in the neuroscience to get large molecules across the BBB would be the use of low-frequency focused ultrasound combined with microbubbles. This non-invasive and reversible technique [154,155] can achieve a transient safe opening of the BBB [155,156]. Successful preclinical studies have already been performed with growth factors, antibodies, genes, viral vectors, and nanoparticles in rodent models of AD and PD [154,156,157]. Recent small clinical studies support the safety and feasibility of this strategy in patients [158]. Further research is needed to determine the safety when the MRI-guided BBB opening is used to improve the delivery of newly developed molecular therapies [156,157].

Furthermore, an upcoming way to improve BBB penetration after parenteral systemic administration is the use of chemical modification or antibody conjugation of native NTFs. Specifically, a covalent modification of NGF with the polyamine putrescine resulted in improved plasma pharmacokinetics and BBB permeability in rats, as compared with native NGF [159]. Moreover, a study by Wu and Pardridge [160] attached biotinylated polyethylene glycol-modified-BDNF to a monoclonal antibody against the transferrin receptor that was linked to streptavidin. This resulted in the ability of the chimeric molecule to bind to the transferrin receptor, which is abundant on brain endothelial cells, and subsequently to undergo receptor-mediated transcytosis through the BBB [160]. Although modification/conjugation strategies are promising for the CNS delivery of peripherally administered NTFs, a major challenge to the clinical implementation of such strategies is the anticipated difficulty in producing large quantities of pharmaceutical-grade preparations and in targeting the products to specific CNS areas [161].

Beyond the BBB permeability, it should be taken into account that, in general, transplanted cells manipulated to (over)express proteins may differentiate into undesirable cell types, with the possibility of tumour formation, risks of host rejection, and inflammation [162,163,164], limiting the widespread use of these manipulated cells, despite their advantages [164]. Viral vector-mediated delivery may already overcome some of the above-mentioned challenges. In particular, virus administration could permanently alter the cells′ ability to make their own NTFs, consequently requiring only a single injection and, thereby, decreasing the invasiveness of the treatment [165]. However, controlling the production of NTF proteins and terminating their expression warrants further research, since cytotoxic effects on host cells and inflammatory responses were seen after the development of self-inactivating viral vectors for in vivo applications [166].

The last delivery method discussed in this review is the application of biomaterials. In general, this method requires a less invasive manipulation with delivery of large amounts of NTFs to the damaged sites. When selecting the delivery method, a number of properties, such as degradability, safety, non-toxicity, and adaptability to release, must be taken into account [167]. Furthermore, biomaterials used for CNS regeneration should be injectable. It should be remembered that natural biomaterials can be immunogenic, but not toxic [167,168]. Synthetic components, on the other hand, do not cause inflammation, but may provoke cytotoxicity [167,169]. A recent technique, which is successfully developed for clinical use in neurodegenerative diseases, includes targeted nano-carriers for recombinant growth factors, therapeutic antibodies, enzymes, synthetic peptides, cell-penetrating peptide-drug conjugates, and RNAi sequences [170]. To enable challenging applications of nano-medicine and precision medicine in the treatment of neurodegenerative diseases, more in-depth research into bio-molecular delivery via nano-carriers for neuronal targeting and repair is needed. According to a recent review by Yu Wu et al., the successful use of macromolecular bio-therapeutics in clinical developments for neuronal regeneration will be aided by recent strategies to improve their bioavailability [170].

It is worth mentioning that many of the challenges discussed above may be overcome by small molecules that target the receptor for the NTF, instead of introducing the NTF itself. The development of small molecule mimetics, with an intrinsic neurotrophic activity and an improved pharmacokinetic profile, is a promising research area. This would allow for specific activation of only one type of receptor, such as TrkA or TrkB and not p75, or vice versa, potentially alleviating the side effects. Interestingly, it has recently been shown that neuro-inflammatory cytokines, such as TNF-α, downregulate both the mRNA and protein levels of TrkA, together with an increase of p75 mRNA expression [171]. This could shift NGF signalling from a neuroprotective to a neurotoxic role, showing that a specific binding of a certain receptor is interesting, especially during pathological (inflammatory) conditions [171]. The use of NTF therapy or NTF mimetics in combination with a TNF-α inhibitor could also be an interesting option. Because several synthetic TNF- α inhibitors induce serious adverse effects in various inflammatory diseases, patients and researchers have recently turned their attention to natural medicines, especially phytochemicals. Phytochemicals targeting TNF- α can significantly improve disease states with fewer side effects, according to the review by Subedi et al. [172]. Several experimental studies have also shown that the administration of bioactive molecules in low doses is effective to obtain pure biological effects with low risk of side effects [153,173].

The discovery and use of peptide mimetics [174] and small molecule ligands for the Trk receptors [175] have attracted considerable interest. Therefore, relatively stable peptide mimetics of NGF have, amongst others, been produced [176]. These analogues may be less immunogenic, more resistant to proteolytic degradation, and able to cross blood–tissue barriers, as compared with their parent molecules. These ligands may be more stable and less expensive to produce than recombinant proteins, and may eventually provide acceptable oral bio-availabilities unattainable with native NTFs. The use of a potent peptide BDNF mimetic that activates TrkB was shown to promote neuronal survival in embryonic sensory neurons of the dorsal root ganglion [177]. Small-molecule BDNF mimetics also have high potency and specificity against TrkB, and can promote neuronal survival, while also inducing differentiation and synaptic function in cultured hippocampal neurons [178]. When administered to mouse models of AD, HD, and PD, the small molecule could rescue cell death to the same extent as the full-length protein BDNF [178]. A number of clinical trials are also currently being conducted with NTF mimetics [175]. Results from these trials, especially in terms of side effects and efficacy, will broaden and improve NTF-based therapy for the treatment of neurodegenerative diseases with acute or chronic neuronal and glial damage.

Although NTF-based therapy has great potential, the greatest uncertainty is whether such an approach by itself is sufficient to halt and reverse the progression of neurodegenerative diseases. Due to the failures of monotherapy in the past, it may be interesting to use combination therapy, instead of the ′single magic bullet′ approach, to address the various disease-causing mechanisms simultaneously. In particular, a combination of several NTFs could be better than using a single NTF for neurodegenerative diseases. For example, studies have shown that BDNF and NT-3, when used in combination, are more effective than either factors alone in increasing the growth of host axons into transplanted spinal cord tissue following spinal cord hemisection in adult rats [33,179]. These synergistic effects may allow combinations of factors to be used at smaller doses than those required of any one factor used alone, diminishing adverse effects and potential for immunogenicity.

Moreover, combination therapy may be particularly useful in the treatment of CNS diseases in which there are multiple neuronal types affected, so that a NTF with maximal activity on a particular cell type can be administered together with another that acts on another cell type. For example, the capacity of NGF to stimulate cholinergic basal forebrain cells is also enhanced by BDNF, which can additionally potently stimulate dopaminergic cells in the midbrain [180]. We can, therefore, envision that a combined use of NTFs may work synergistically to restore neuronal function.

Besides NTFs, a number of other biological agents have emerged that show regenerative properties in neurodegenerative diseases, such as vascular endothelial growth factor (VEGF) [181,182,183,184,185,186,187,188], insulin-like growth factors (IGFs) [189,190,191,192,193,194,195,196,197,198,199,200,201,202,203], the cellular communication network (CCN) family [204,205,206,207,208], and erythropoietin (EPO) [209,210,211,212,213,214,215,216,217,218,219,220,221,222,223,224,225,226,227,228,229,230,231,232,233,234], with varying, but also promising results.

## 6. Conclusions

To date, several NTF distribution vectors and systems have been applied to deliver exogenous NTFs into the CNS, with variable results. In most cases, the translational capacity from bench to bedside was limited. The challenges that currently need to be overcome include the amount of NTFs released, BBB permeability if administered peripherally, the invasiveness of the delivery route, the half-life of the vehicle, and the occurrence of possible side effects. The combination of all these challenges is probably the reason why the application of NTFs has, so far, not been effective for the long-term regeneration of target tissues, especially in the brain. In addition, beyond the use of a single NTF, combination therapies, targeting multiple pathways or using smaller molecules, such as NTF mimetics, would be a more effective treatment option in neurodegenerative diseases. Nevertheless, it is important to continue research into the optimization of cellular-, viral vector-, and biomaterial systems to provide standards for clinical applications.

## Figures and Tables

**Figure 1 ijms-24-03866-f001:**
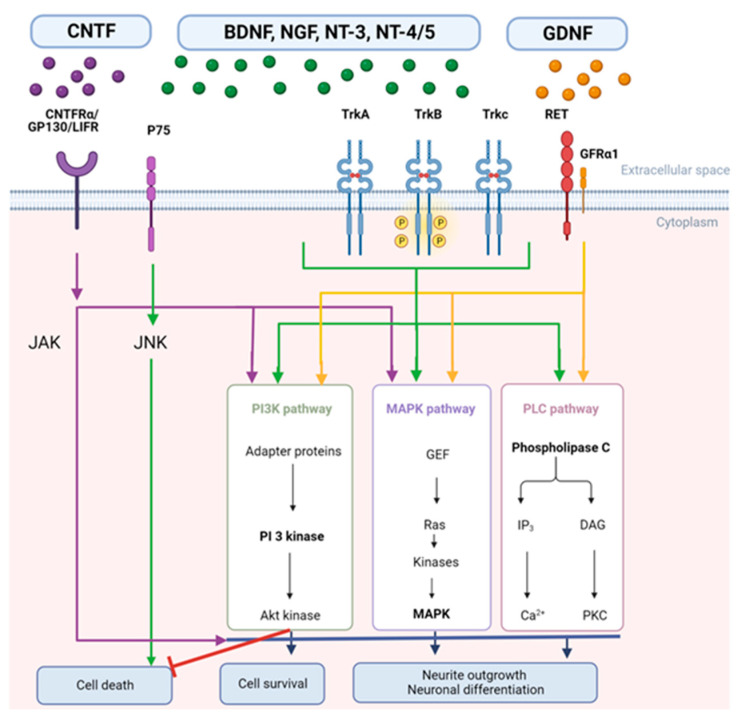
The different NTF signalling pathways. The colours of the NTFs correlate with the colours of the arrows. Abbreviations used: Phospholipase C-γ (PLC-γ), phosphoinositide-3-kinase (PI3K), mitogen-activated protein kinase (MAPK), leukaemia inhibitory factor (LIFRβ), glycoprotein 130 (GP130), Janus kinase/signal transducer and activator of transcription (JAK), c-Jun N-terminal kinase (JNK), GDNF family receptor alpha-1 (GFRα1), Inositol trisphosphate (IP3), diacylglycerol (DAG), protein kinase C (PKC), Guanine nucleotide exchange factor (GEF) (Created with BioRender.com; adapted from Pietrucha-Dutczak et al. [76] and Kashyap et al. [121]).

**Figure 2 ijms-24-03866-f002:**
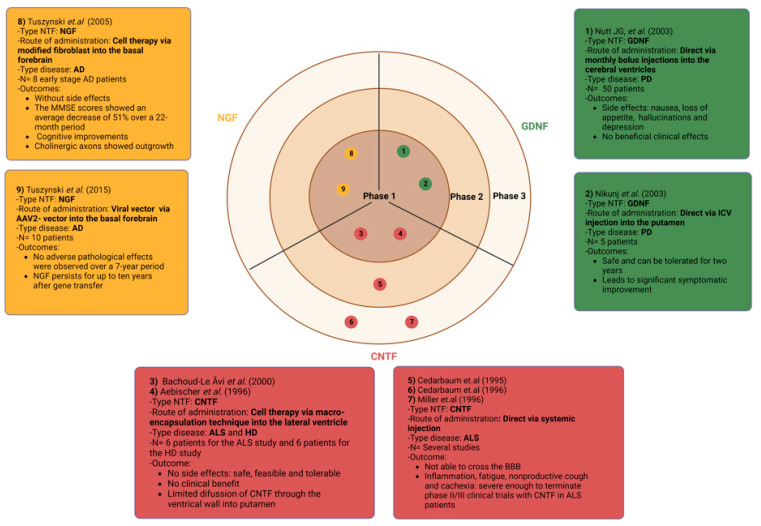
A summary of the current status of clinical trials applying neurotrophic factors. Several studies with NTFs for neurodegenerative diseases are still in the preclinical phase, whereas some of the clinical trials already initiated were terminated due to side effects or no clinical improvement. The different colours of the boxes correlate with the colours given to the NTF. Abbreviations used: glial cell-derived neurotrophic factor (GDNF), ciliary neurotrophic factor (CNTF), nerve growth factor (NGF), Alzheimer’s disease (AD), Parkinson’s disease (PD), Huntington’s disease (HD), amyotrophic lateral sclerosis (ALS), blood-brain barrier (BBB), Mini-Mental Status Examination (MMSE) scores, and N= enrolled patients (Created with BioRender.com) [59,60,61,63,64,65,66,67,69,70].

## Data Availability

Not applicable.

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
