# Peer review of "Neurotrophic Factors as Regenerative Therapy for Neurodegenerative Diseases: Current Status, Challenges and Future Perspectives"

_ijms, 2023, doi:10.3390/ijms24043866_

Round 1

Reviewer 1 Report

The Review entitled “Neurotrophic factors as regenerative therapy for neurological diseases: current status, challenges and future perspectives” by El Ouaamari et al. describes the current state of knowledge, challenges and future perspectives of NTFs and their regenerative effects in chronic inflammatory and degenerative disorders.

In my own view, I see little efforts from the Authors to make an inspiring review of the current knowledge and future perspective in this very hot topic. By the way, only about 3% of the quoted citations are from the last two years, while the overwhelming majority of the cited sources are quite seasoned. Moreover, sometimes information regarding year of publication and/or even Journal name is missing. There is no efforts in making worthwhile schemas and/or illustrations to make the reading more attractive and impactful. Given these limitations I cannot consider this Ms. worthy of publication, at least, in its current form.

Reviewer 2 Report

The role of NTFS in neurodegenerative disorders is very well studied and several of them have gone through clinical trials. The title Neurotrophic factors as a regenerative therapy for neurological 2 diseases: current status, challenges and future perspectives suggest this review article focuses on the functions of these NTFs, how NTFs can be used as therapy, and what are the limitations of delivering NTFs in neurological disorders. 

However, the Authors mentioned neurological disorders in the title but used the term neurodegenerative disorders in every section of the article. Authors should reconsider changing the title using neurodegenerative disorders.

Although the authors wrote the delivery techniques of the NTFs, it lacks an elaborated function of these NTFs in NDs which can be added in a separate section as Section 2.  

Section 2. Result should be changed to something like “Delivery of NTF and associated challenges”.

Authors should also include promising alternatives or mimetics of NTFs.

Moreover, it lacks enough figures and tables to make the article more informative.

Overall, the authors should also reconsider rewriting the manuscript as the current form does not give a clear idea of NDs, their pathogenesis, and how NTFS is linked to the pathogenesis (mechanisms) of the disease progression. The article is not well structured. This article can not be published in its current form. 

Reviewer 3 Report

Dear Authors, 

The review article titled “Neurotrophic factors as regenerative therapy for neurological diseases: current status, challenges and future perspectives” clearly describes the potential role of neurotrophic factors against different neurological disorders, focusing on different strategies in order to target the damaged areas and the delivery methods, taking also into account the main molecular pathways involved as reported and summarized in figure 1.

However, in my opinion, minor revision are need in order to improve the quality of your paper and to better discuss this fascinating and promising field of research.

1- Page 3, lines 123-125: this sentence should be rephrased since it appears that basal forebrain is a part of the peripheral nervous system. I suggest the following: "In the peripheral nervous system (PNS), NGF is the dominant NTF, acting on sympathetic and sensory neurons. In the CNS, NGF specifically provides trophic support to cholinergic neurons of the basal forebrain (BFCNs) that express TrkA (Figure 1)."

2-Figure 1: under the white three windows there is a blue line following the purple one on the left side. Please explain the different colour meaning.

3- The Authors correctly mention the main pathways involved by neurotrophic factors such as NGF and BDNF (as also reported in figure 1). However, a mention on one of the first paper on this topic (dot: 10.1074/jbc.M114.587188) should be done.

4- It has been recently largely reviewed that NGF has a double role both in anti- and pro-inflammatory response. Furthermore, cytokines such as IL-1β, TNF-α and IL-6  (doi: 10.3390/ijms18051028 ) induce the NGF over expression, such as the CNTF (doi: 10.4049/jimmunol.165.4.2232).

5- Moreover, concerning the NGF molecular pathway and different receptors, the Authors correctly discuss, even if very little, the need to activate only one NGF receptor such as trkA and B and not p75 (page 11, lines 502-504). This is true, since it has been recently reported that neuroinflammatory cytokines, such as TNFα, downregulates both mRNA and protein levels of the tropomyosin receptor kinase A (TrkA), together with an increase of p75 mRNA expression (doi: 10.3390/ijms21176128 ). Thus, since NGF shift between a neuroprotective and neurotoxic role by different receptors, this paper might help Authors in reinforcing their hypothesis to allow the activation of just one receptor especially during pathological (inflammatory) condition.

6- The Authors correctly mention that the “single magic bullet approach” is not truthful. On this regard, since NGF and other neurotrophic factors have a putative role in neurodegenerative disease, the Authors should also mention the potential role of phytochemical (such as those reported effective against TNF-α induced neuroinflammation) (doi: 10.3390/ijms21030764) in order to counteract, for example, the neuroinflammation in order to increase the NTF effects.

7-Finally, concerning the different delivery methods, the Authors should also mention the ultrasound application for blood-brain barrier opening in order to deliver the neurotrophic factors such as BDNF and GDNF and others (doi: 10.1038/s41598-019-55294-5; doi: 10.3389/fbioe.2022.961728; doi: 10.1002/mds.27804), as recently reported in neuroscience field.

8- Page 11, line 486: The term “neurotrophic factors” should be abbreviated as “NTFs”.

9- Page 11, line 490: “Wu and Pardridge” without “et al.” since the quoted paper has only two Authors. 

Author Response

"Please see the attachment

Reviewer 4 Report

As interesting as the topic covered is, I think some changes need to be made to the bibliography: most of the studies reviewed are more than a decade old; several more recent studies should be included. For example, the following studies about BDNF should be included: Molinari, Claudio, et al. "The role of BDNF on aging-modulation markers." Brain Sciences 10.5 (2020): 285. and Wu, Yu, et al. "Self-Assembled Nanoscale Materials for Neuronal Regeneration: A Focus on BDNF Protein and Nucleic Acid Biotherapeutic Delivery." Nanomaterials 12.13 (2022): 2267. 

Also in the introduction and discussion the references should be better included, in some sentences there are too many (in line 71 alone almost a hundred are listed), I suggest selecting only those with greater impact.

Finally, I would suggest making the references uniform.

Round 2

Reviewer 1 Report

IJMS# 1956641 El Ouaamari Y et al.

I appreciated the efforts by AA in revising the Ms. Updated citations, re-organization of the text have certainly improved the draft. The new Figure would summarize somewhat the current status of studies in which NFTs were administered. Unfortunately, the Figure is not self-explanatory for the readership. Conceptually, the first type of the studies is preclinical studies (inner position), followed by phase 1, phase 2 and 3 studies. Furthermore, some important informations (i.e., type of NTF, way of delivery, adverse events, type of disease, n=enrolled patients, outcomes) should be reported in a clearer way. I encourage the AA to improve the Figure and/or add a further Table/scheme.

From my point of view the revised Ms. needs to be improved for the acceptance and publication. For this reason, I recommend a major revision.

In particular:

1) The text style should be improved. The manuscript needs an extensive revision for language and grammar (for example see lines #137-151, #177-187, #550-557, #591-595, etc…).

2) Some statements appear incomplete (lines #404-406 “Although, there are some disadvantages of using AV vectors, namely immunogenicity, replicability and small insertion size of the vectors.” ). The AA should revise the text.

3) Figure 2 and figure legend. In my opinion, the Figure is not sufficient clear to the readership (see general comments). What is the significance of different colours (i.e., blue, green, red and yellow)? What is the significance of the number into the coloured circles? The AA should explain the significance in the text and figure legend.

Minor comments

-“L’Hermitte’s sign” should be replaced with "Lhermitte sign"

- line #289 the “….amyloid β (Aβ)…” should be moved from this line to line #288. Please check it in the text.

Reviewer 2 Report

The author has addressed all the comments well.

Thus it can be accepted.

Author Response

Thank you for all your suggestions, it certainly improved the review. 

Reviewer 4 Report

I would like to thank you for taking the time and effort to proofread the manuscript. Considering the changes made to the manuscript I believe that the work is improved. 

Author Response

We thank you for making the suggestions. These have certainly improved the review. 

Round 3

Reviewer 1 Report

IJMS# 1956641 El Ouaamari Y et al.

I appreciated the efforts of the AA in order to improve the revised Ms. Some statements are revised and  style text improved. Unfortunately, I was not able to follow all the corrections made by the AA given that these are not displayed as comments in the file. Furthermore, the corresponding text from line #136 to line #184 is missing in the revised PDF file.

Nevertheless, I continue to see some minor grammar errors (lines #202-203 “….Apart from GDNF and CNTF, other known factors are the neurotrophins. These  group consist of four members that share a common ancestral gene, as well as being similar in structure….”. Please check and replace the second phrase with …“This group consists of…”.

The new version of Figure 2 and corresponding legend of the Figure are certainly improved.

Minor grammar errors are still observed. Please check the text in the boxes. “Outcome” should replaced with “Outcomes” and “…NGT persist for up to ten..” should replace with “NGT persists for up to ten..”

In order to check all the text corrections, I would suggest the AA to re-upload the PDF file. For this reason, I recommend a last minor revision of the Manuscript.
